# Analysis of vehicle carbon emission characteristics on expressways in mountainous plateau areas based on the coupled simulation of CarSim/TruckSim and MOVES

**Jianping Gao**[1], **Xin Huang**[1], **Yunyong He**[2]\*, **Enhuai He**[2], **Lu Sun**[2], **Changfeng Yang**[2]

**1** School of Civil Engineering, Chongqing Jiaotong University, Chongqing, China, **2** Sichuan Highway Planning, Survey, Design and Research Institute ltd., Chengdu, China

\* heyunyong@schdri.com

## Abstract

Expressways in mountainous plateau areas exhibit complex driving conditions and harsh climatic characteristics that continuously impact vehicle carbon emissions throughout their entire lifecycle and determine the carbon emission levels of expressways during the operational period. To study the carbon emission characteristics of expressways in the western Sichuan Plateau mountainous area, a coupling simulation analysis method combining Car-Sim/TruckSim simulation software and the Motor Vehicle Emissions Simulator (MOVES) was employed to analyze the spatial distribution characteristics of vehicle-specific power (VSP). This analysis was based on the alignment data and operational environment of the Wenchuan–Barkam Expressway in the mountainous area of the western Sichuan Plateau. A conversion formula was established to calculate the cumulative VSP per second and total carbon emissions of each unit. Statistical analysis was conducted on the distribution of carbon emissions equivalent per kilometer across the different alignment units. Distribution ranges of carbon emissions equivalent per kilometer for light-duty vehicles (LDVs) and heavy-duty vehicles (HDVs) at each alignment unit were 60–240 g and 100–1600 g, respectively. The grey relational analysis method was used to quantify the relationship between the carbon emissions equivalent per kilometer of alignment units, circular curves radius, and average grade. Based on the vertical variations in climatic changes with altitude in the mountainous plateau area, a comparative analysis was performed on the trends in the effects of altitude, season, and vehicle starting frequency on carbon emissions. Frequent vehicle starts significantly impacted carbon emissions, and this impact was significantly higher in winter than in summer. Carbon emissions equivalent of the LDV and HDV starting twice on average in summer were approximately 1.09- and 1.04-times higher, respectively, than that when vehicles were started 0.5 times; whereas, in winter, they were 1.17- and 1.07-times higher, respectively.

**Data Availability Statement:** All relevant data are within the manuscript and its Supporting Information files.

**Funding:** This work was supported by the Transportation Technology Project of Sichuan Provincial [Grant numbers 2023-A-08]; the Transportation Technology Project of Sichuan Provincial [Grant numbers 2022-A-07]; and Chongqing Postgraduate Joint Training Base [Grant numbers JDLHPYJD 2020015].The funder had no role in study design, data collection and analysis, decision to publish, or preparation of the manuscript.

## Introduction

In the context of global warming, the climate in plateau mountainous areas demonstrates a notable warming trend. Climate change at various scales occurs earlier in these areas than in other regions, and the magnitude of change is more significant [1,2]. From 1960–2010, the rate of temperature increase on the Qinghai–Tibet Plateau was 0.2°C per decade, and that in the alpine region of the Qinghai–Tibet Plateau was 0.29°C per decade [3]. The temperature increase has resulted in permafrost thawing, retreating glaciers, and an increase in extreme climate events on the Qinghai–Tibet Plateau, all of which have profoundly impacted the ecological environment on the plateau. The fragility of this environment necessitates greater attention to carbon emission control in plateaus and mountainous areas [4]. Highway alignment designs directly influence vehicle carbon emissions and fundamentally determine the overall level of highway carbon emissions. Therefore, research on vehicle carbon emission characteristics on expressways in mountainous plateau areas and analyzing the distribution pattern of carbon emissions along alignment units during the free-flow state of a single vehicle is beneficial for elucidating the relationship between low-carbon emissions and alignment design indicators for highways in mountainous plateau areas. Such research provides a theoretical foundation for low-carbon alignment design in these areas.

Carbon emissions from vehicles are closely related to operating conditions and environments characteristics. Compared to the traditional speed–acceleration carbon emission correlation model, vehicle-specific power (VSP), which was proposed by Jimenez–Palacios [5], provides an improved correlation between vehicle carbon emissions and operating conditions. Qi et al. [6] conducted real vehicle tests on light-duty vehicles (LDVs) and heavy-duty vehicles (HDVs) on the Beijing–Shanghai Expressway, analyzed the VSP distribution characteristics across different speed intervals, and concluded that the distribution was nearly a normal distribution. Hong et al. [7] utilized the China automotive test cycle method to collect driving condition data for LDVs on urban roads and established a framework for calibrating vehicle emissions based on these urban LDV driving conditions. Holmén et al. [8] collected exhaust pollutants produced by different drivers under specific traffic and environmental conditions, and proposed that vehicle acceleration is the main factor affecting exhaust emissions. Suarez et al. [9] recorded actual driving data from 20 drivers and concluded that $CO_2$ emissions varied by approximately 5% between the most and least favorable driving behaviors. Barth et al. [10] emphasized that vehicle operating speed is the most significant factor influencing carbon emissions, proposed a "U shape" curve to describe the relationship between carbon emissions and vehicle speed, and constructed a quadratic parabolic model that relates speed to carbon emissions. In terms of the impact of the environment characteristics on vehicle carbon emissions, some studies have found that $CO_2$ emissions significantly increase under both high- and low-temperature conditions [11,12]. At an ambient temperature of -20°C, the $CO_2$ concentration during idling exceeds 16% [13]. Due to the decline in exhaust gas recirculation and post-treatment performance, $NO_X$ emissions at ambient temperatures of 0–5°C are 82–192% higher than those at 15–20°C [14,15]. Furthermore, as relative humidity increases from 10% to 40%, the $CO_2$ emission rate for LDVs decreases from 4.2 g/s to 3.6 g/s [16]. Real vehicle tests have shown that altitude significantly impacts $CO_2$ emissions, and that within the elevation range of 2270–4540 m, the $CO_2$ emission factor for light gasoline vehicles ranges from 161.66–181.98 g/km [17].

In a study on the impact of road alignment conditions on vehicle carbon emissions, Dong et al. [18] analyzed the influence of the circular curve radius and concluded that the minimum circular curve radius affecting vehicle carbon emissions was 500 m. Jiao et al. [19] proposed that grade is the most critical factor affecting vehicle $CO_2$ emissions, and a strong correlation exists between grade and carbon emissions [20]. Carbon emissions increase significantly at grades > 3% [21]. Ko et al. [22] simulated various driving states of vehicle acceleration and

deceleration using the Motor Vehicle Emissions Simulator (MOVES) to evaluate vehicle energy consumption and carbon emissions. The authors found that on a 3-km sloped road section, when the vehicle speed decreased by $> 20$ km/h, fuel consumption was five times higher than that with a speed reduction of $\leq 10$ km/h. Subsequently, the authors analyzed the impact of highway vertical curve design on vehicle fuel consumption and emissions and exemplified that a gentle curvature design of vertical curves offers both environmental and economic benefits over the entire lifespan of a highway [23].

The mountainous plateau areas of western Sichuan exhibit complex topography and extreme altitude variations. The proportion of curved sections on expressways in this area is high, leading to frequent acceleration and deceleration of vehicles. In upslope sections, slope resistance can cause a sharp increase in vehicle power demand. In addition, The harsh climate in plateau mountainous areas, characterized by low pressure, low oxygen, and low temperature, have a complex impact on vehicle carbon emissions. Low pressure and oxygen reduce engine intake, causing incomplete combustion. Low temperatures lead to quick heat dissipation, poor fuel atomization, and reduced combustion efficiency, all of which increase vehicle carbon emissions. Domestic and international research on vehicle carbon emission characteristics primarily relies on factors such as driving behavior and road grades, with most studies focusing on urban roads and general highways, and there has been relatively limited research on vehicle carbon emissions from expressways in mountainous plateau areas. Highway alignment indicators partially determine vehicle operating conditions, and the continuity of the alignment significantly influences road carbon emission levels [24,25]. Previous studies have mainly focused on the emission characteristics of vehicles under specific operating conditions, and it is hard to directly apply them to guide alignment design. We aim to conduct CarSim and TruckSim dynamic simulation tests on LDVs and HDVs on expressways in mountainous plateau areas. By doing so, we can obtain the vehicle operating conditions and establish a MOVES vehicle carbon emission model under the plateau environment with low pressure and low oxygen. Then, we will analyze the characteristics of road alignment conditions and environmental features that influence vehicle carbon emissions. Our research contributes to the literature on the design, planning and management of expressways in mountainous plateau areas in the following aspects: 1) We divide the alignment analysis units of expressways in mountainous plateau areas and reveal the spatial difference characteristics of the VSP distribution of LDVs and HDVs in different alignment units; 2) We analyze the correlation between the total carbon emissions of different alignment units and the cumulative sum of VSP, construct a carbon emission calculation model based on the cumulative sum of VSP of alignment units, and realize the effective evaluation of vehicle carbon emissions in the design stage of expressways in mountainous plateau areas; 3) We analyze the distribution characteristics of vehicle carbon emission factors in different alignment units of expressways in mountainous plateau areas and quantify the effect degree of key alignment indicators on vehicle carbon emissions, providing fundamental support for the low-carbon alignment design of expressways in mountainous plateau areas. 4) We construct a localized MOVES vehicle carbon emission model for expressways in mountainous plateau areas and focus on analyzing the influence law of the difference characteristics of temperature and humidity in mountainous plateau areas with changes in altitude and season on vehicle carbon emissions.

## Methods

### Dynamic simulation

We used CarSim and TruckSim simulation software to simulate actual stress states and vehicle operating conditions, and developed dynamic simulation models for various vehicle types on

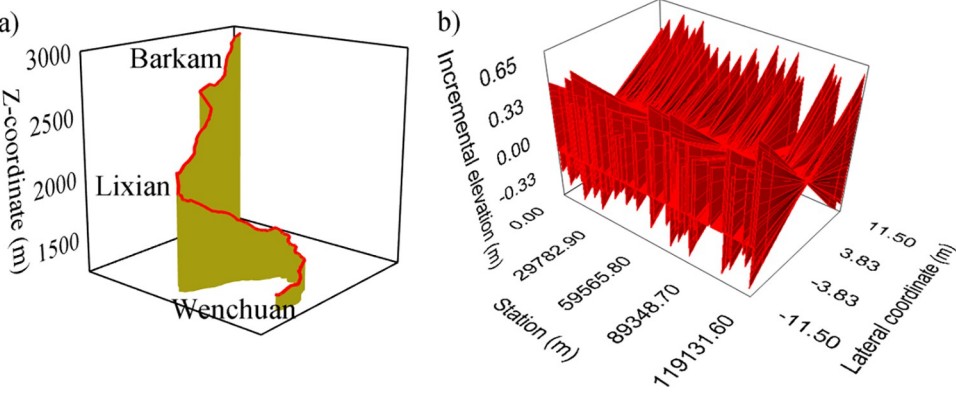

**Fig 1. Road geometry.** (a) Road alignment. (b) Road superelevation.

an expressway in the mountainous plateau areas of western Sichuan, i.e., the Wenchuan–Barkam Expressway (Wenma Expressway). The model considers the driving force, braking force, rolling resistance, air resistance, tire lateral force, the self-weight of the vehicle, supporting force, centrifugal force, and inertial force that the vehicle experiences during the driving process. Moreover, based on formulating the target speed curve, it simulates and analyzes the vehicle's operating conditions, such as the acceleration, speed, and VSP, on a second-by-second basis.

## Road model

The Wenma Expressway is located in the interlaced contact zone between the Qinghai-Tibet Plateau's eastern margin and the Sichuan Basin's northwestern edge, and the route altitude increases from 1300 m to 3000 m. The design speed is 80 km/h. The simulated section runs from Wenchuan through Lixian to Barkam, covering a distance of approximately 119 km. The minimum circular curve radius along the route is 405 m, and curved segments account for approximately 60% of the total length. There is an average of 0.982 intersections per kilometer. The maximum route grade is 4%, and the average frequency of grade changes per kilometer is 0.806. The Wenchuan to Lixian section is dominated by medium and low mountains and river valley terraces, with relatively wide valleys, an average grade of 1%, and a distance of approximately 64 km. The Lixian to Barkam section mainly features alpine and gorge landforms, with steep mountains and deep valleys, an average grade of 1.76%, and a distance of about 55 km. The cross-section of the road has two-way four lanes, and the pavement type is asphalt concrete. The road geometry is illustrated in Fig 1 and the distribution of road alignment indicators is shown in Fig 2.

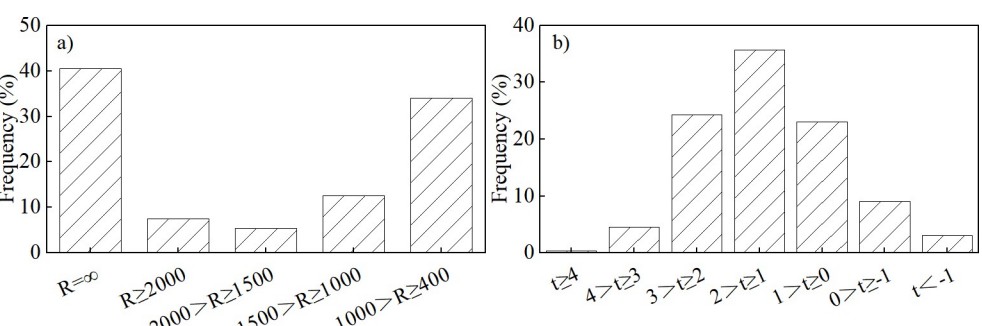

**Fig 2. Distribution of road alignment indicators.** (a) Radius (R) of circular curve (m). (b) Grade (t), vertical rise/slope length (%).

**Table 1. Simulation parameters of vehicle.**

| | Vehicle types | Engine power (kw) | Drive type | Sprung mass (t) | Unsprung mass (t) | Load mass (t) | Fuel types |
|---|---|---|---|---|---|---|---|
| LDV | passenger car | 125 | front-wheel drive | 1.27 | 0.142 | — | gasoline |
| HDV | cargo truck (four-axle) | 300 | 8×4 (rear-wheel drive) | 4.457 | 2.12 | 7.5 | diesel |

## Vehicle model

An LDV and HDV were selected as simulation subjects to analyze the impact of different vehicle and fuel consumption types on vehicle carbon emissions. According to the traffic composition data of the Wenma Expressway in the past four years, the vehicle types with the largest proportions among LDVs and HDVs are passenger cars and cargo trucks (four-axle), respectively. From 2020 to 2023, the proportion of passenger cars in LDVs was 92.41%, 92.4%, 92.39%, and 92.41% year by year, while the proportion of cargo trucks (four-axle) in HDVs was 39.81%, 39.43%, 39.28%, and 39.25% year by year. Accordingly, passenger cars and cargo trucks (four-axles) were selected as the representative vehicle types for LDVs and HDVs. The external dimensions of the vehicles were referenced from the "Technical Standards for Highway Engineering (JTG B01-2014)" [26], and gasoline and diesel fuels were utilized. Vehicle model parameters were derived from the CarSim and TruckSim databases, with their mathematical models based on measured data provided by vehicle manufacturers and calibrated through vehicle bench tests to ensure high reliability. The corresponding technical parameters were revised considering differences in vehicle dynamic performance, both domestically and internationally. The simulation parameters are presented in Table 1.

## Operating speed prediction

The operating speed prediction model from the "Specification for Highway Safety Audit (JTJ B05-2015)" [27] was adopted. The vehicle operating speed was calculated and used as the target speed curve for vehicle dynamics simulation by setting parameters such as the expected speed, minimum speed, and acceleration, thereby enhancing the universality and representativeness of the simulation test. The prediction model divides the alignment analysis units based on the highway alignment conditions (Table 2). Horizontal curve and curve–slope combination sections were classified into two alignment units at the midpoint of the curves and categorized into four types according to the connection mode of the front and rear alignments: exit-to-curve, exit-to-straight, curved entrance, and straight entrance. The start and end points of these alignment units served as characteristic points for calculating the operating speed.

The target speed curves for both LDV and HDV were established, and the actual operating speeds of the vehicles during the dynamic simulation were obtained, as shown in Fig 3. The maximum altitude of the simulated section was 2900 m; therefore, the effects of high altitudes ($> 3000$ m) on operating speed were not considered.

**Table 2. Classification standards for expressway alignment analysis units.**

| Vertical alignment | Horizontal alignment | |
|---|---|---|
| | R > 1000 m | R ≤ 1000 m |
| t < 3% | L >200 m, straight sections<br>L ≤ 200 m, short straight sections | horizontal curve sections |
| t ≥ 3% | longitudinal slope sections | curve–slope combination sections |

L, length of alignment units (m).

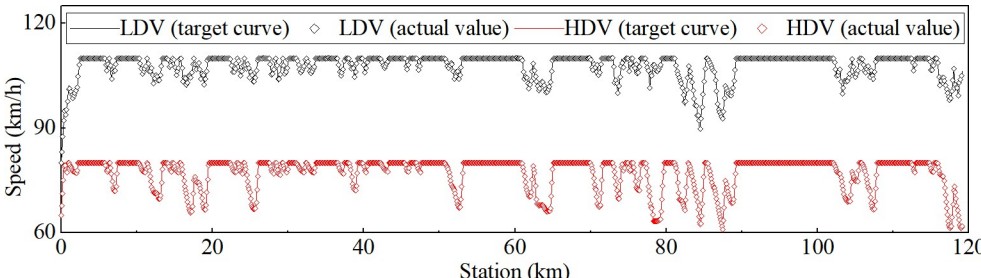

**Fig 3. Target curves and actual value of the vehicle operating speed.**

## Carbon emission simulation

MOVES is a carbon emission assessment model developed by the United States Environmental Protection Agency based on vehicle driving speed and VSP [28]. The model was utilized to set parameters such as vehicle operating conditions, vehicle age, fuel type, ambient temperature and humidity, section length, and vehicle type to analyze carbon emission characteristics during vehicle operation on expressways in mountainous plateau areas. The formula for calculating vehicle carbon emissions is presented in Eq 1 [29].

$$TE_{\text{process sourcetype}} = \left( \sum ER_{\text{process,bin}} \times Ac_{\text{bin}} \right) \times Aj_{\text{process}} \tag{1}$$

Here, $TE_{\text{process sourcetype}}$ is total emissions (g); $ER_{\text{process,bin}}$ is the emission rate (g/s); $Ac_{\text{bin}}$ is activity, and; $Aj_{\text{process}}$ is adjustment.

Carbon emission limit standards and geographical conditions in the United States differ from those in China. Consequently, selecting the model parameters required localization. The simulation time was based on 2015 (when the fuel consumption standards in China and the US were comparable), and the parameters about vehicle age were determined accordingly. Other MOVES model parameters are detailed in Table 3.

## Results and discussion

### Spatial distribution of VSP

**VSP distribution along the road.**  VSP characterizes the engine's power output when a vehicle overcomes slope, rolling, and air resistance and maintains the vehicle's operating state per unit mass (1 t). By simulating the vehicle's driving conditions at the target speed, the VSP of the LDV and HDV was output second-by-second (Fig 4), and the VSP distribution patterns of both vehicles were statistically analyzed (Fig 5).

The results showed that the VSP change trends for the LDV and HDV in the simulated section were similar, and the VSP distribution patterns were both close to a normal distribution. The VSP distribution range of the LDV was broader than that of the HDV. The mean VSP value for the LDV (13.7 kW/t) was approximately twice that for the HDV (6.5 kW/t), and the proportion of operating time when the VSP was at the mean value was lower for the LDV than for the HDV. Additionally, in the Lixian to Barkam section (K64–K119), the average grade

**Table 3. Model parameter of MOVES.**

|  | Location | Ambient temperature (°C) | Relative humidity (%) | Altitude (m) | Vehicle types | Fuel types | Road types |
|---|---|---|---|---|---|---|---|
| Wenma Expressway | Lixian | 1 | 60 | 1900 | LDV/HDV | gasoline /diesel | expressway |
| Model parameter | Danver | -2.5 | 57.2 | 1790 |  |  |  |

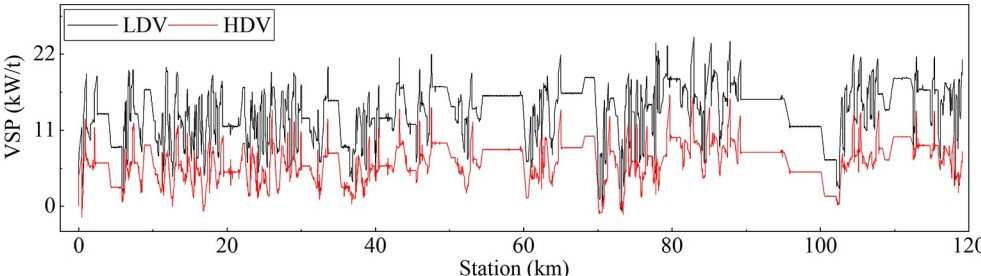

**Fig 4. VSP variation curve.**

increases from 1% to 1.76%, compared with the Wenchuan to Lixian section (K0–K64). The mean VSP value for the LDV increased from 12.57 kW/t to 14.79 kW/t, and that for the HDV increased from 5.22 kW/t to 6.93 kW/t. Obviously, under the same grade change, the mean VSP value growth rate for the HDV (33%) is higher than that for the LDV (18%).

## VSP distribution in different alignment units

Vehicle operating time and section mileage of different alignment units were determined according to the division criteria of the alignment analysis units in the expressway operating speed prediction model (Table 2). The mileage of straight sections of the Wenma Expressway accounted for the highest percentage (61.56%), that of the horizontal curve sections ranked second (32.76%), and that of the curve–slope combination, short straight, and longitudinal slope sections was relatively low (5.68%).

VSP is closely related to vehicle speed, acceleration, curve radius, and grade. To investigate the VSP spatial distribution characteristics on expressways in mountainous plateau areas and analyze the distribution status of VSP within various alignment units, we statistically analyzed the VSP data recorded every second across 11 distinct alignment unit categories: horizontal curve sections (exit-to-curve); horizontal curve sections (exit-to-straight); horizontal curve sections (curved entrance); horizontal curve sections (straight entrance); curve–slope combination sections (exit-to-curve); curve–slope combination sections (exit-to-straight); curve–slope combination sections (curved entrance); curve–slope combination sections (straight entrance); straight sections; short straight sections, and; longitudinal slope sections (Fig 6).

Statistical analysis revealed that the VSP distribution for the LDV and HDV was notably broad in the horizontal curve and straight sections, with a higher concentration in the curve–slope combination and longitudinal slope sections and a more even distribution in the short

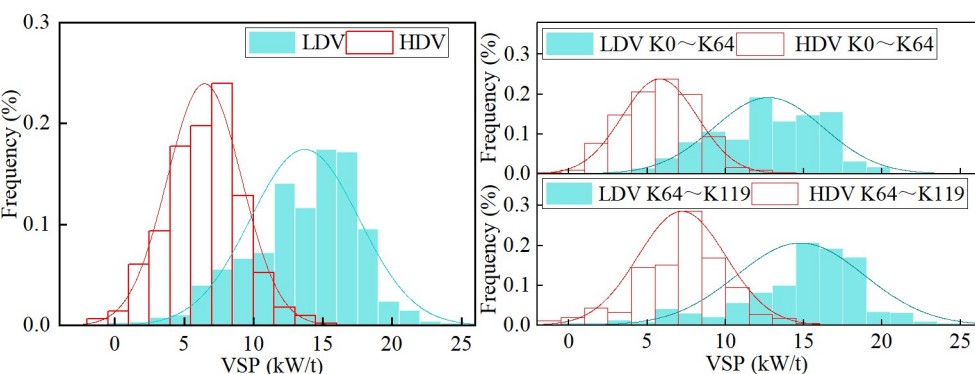

**Fig 5. VSP distribution patterns.**

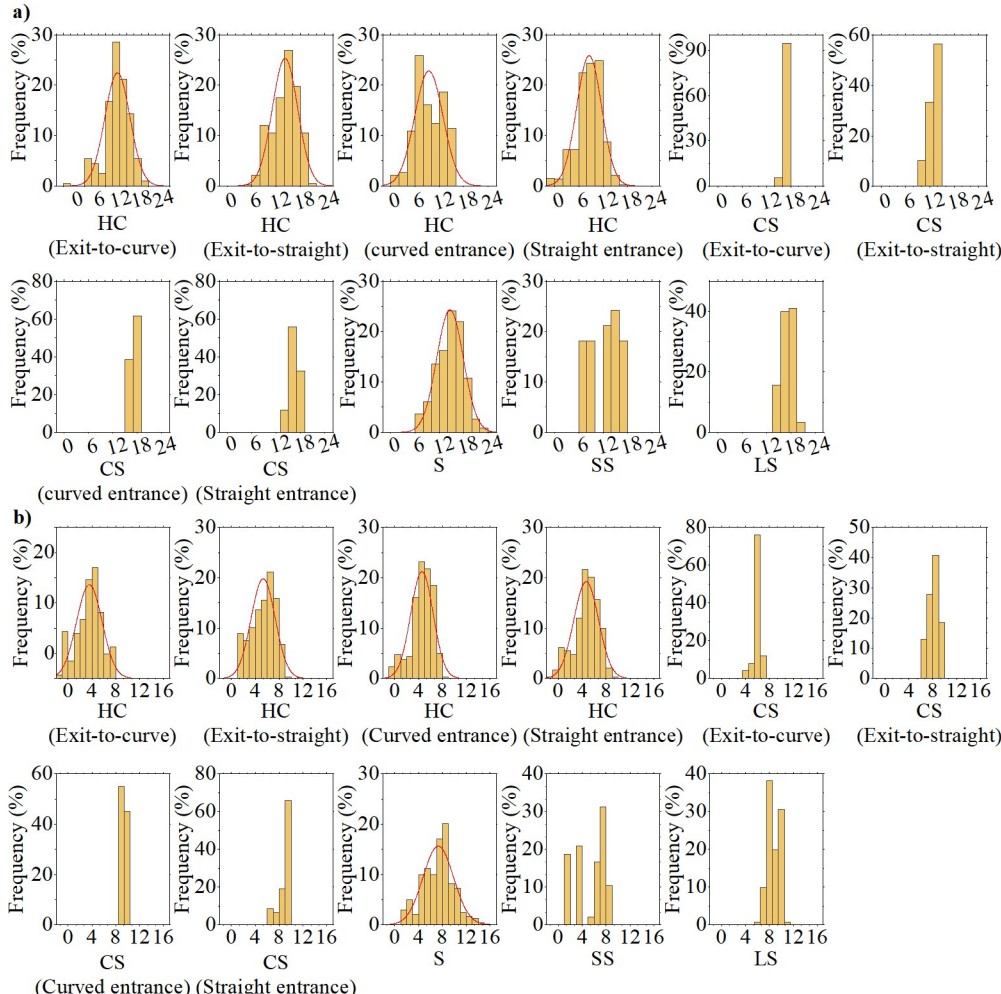

**Fig 6.** Spatial distribution of VSP from the (a) LDV and (b) HDV. HC, horizontal curve; CS, curve–slope combination; S, straight; SS, short straight; LS, longitudinal slope.

straight sections. We calculated the mean VSP value for the LDV and HDV in each unit of the HC (exit-to-curve), HC (exit-to-straight), HC (curved entrance), HC (straight entrance), and straight sections. We conducted Kolmogorov–Smirnov (K–S) and Smirnov–Wolfowitz (S–W) tests to evaluate the normality of the mean VSP value distribution patterns within the same alignment unit type. The K–S test was used for sample sizes > 50, and the S–W test was applied for those < 50. The outcomes of these tests and the parameters indicating a normal distribution are presented in Table 4. The mean VSP value distributions for alignment units in the horizontal curve and straight sections followed a normal distribution, with significance > 0.05.

To visually analyze the spatial distribution patterns of VSP for LDV and HDV across various alignment units, box plots depicting VSP spatial distributions are presented in Fig 7.

**Table 4. Normality test for VSP mean value distribution of different alignment units.**

| Alignment units | HC (exit-to-curve) | | HC (exit-to-straight) | | HC (curved entrance) | | HC (straight entrance) | | Straight | |
|---|---|---|---|---|---|---|---|---|---|---|
| Testing method | K-S | S-W | K-S | S-W | K-S | S-W | K-S | S-W | K-S | S-W |
| LDV | 0.76 | 0.314 | 1 | 0.228 | 0.92 | 0.482 | 0.85 | 0.235 | 0.28 | 0.04 |
| HDV | 0.47 | 0.663 | 0.82 | 0.189 | 0.83 | 0.089 | 0.45 | 0.051 | 0.24 | 0.021 |

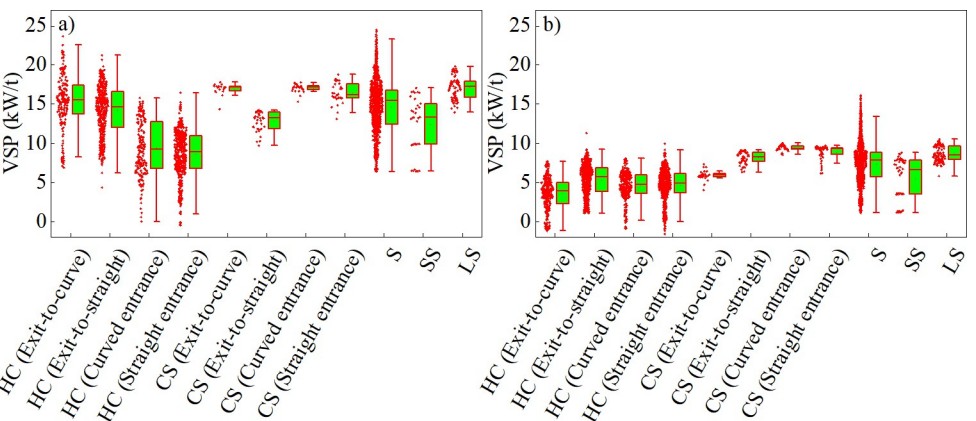

**Fig 7.** Box plots of VSP spatial distribution of the (a) LDV and (b) HDV.

The findings indicated the following:

(1) The VSP distribution range for the LDV and HDV was most extensive in straight sections and narrowed significantly in the CS (curved entrance). The mean VSP value was highest on sections with grades ≥ 3%, (i.e., 17.2 kW/t for the LDV and 8.7 kW/t for the HDV). Notably, the largest difference in the mean VSP value between the two vehicle types was in the HC (exit-to-curve), at 11.7 kW/t, and the smallest difference was in the HC (straight entrance), at 3.9 kW/t.

(2) Across straight, short straight, and longitudinal slope sections, the ranking of the mean VSP value for the LDV and HDV was consistently longitudinal slope section > straight section > short straight section.

(3) There was a pronounced disparity in the mean VSP value between the exit and entrance sections of the horizontal curve for the LDV, with the exit section being approximately 1.65-times higher than that of the entrance section. In contrast, there was a relatively minor difference for the HDV. The LDV exhibited a significant difference in the mean VSP value between the curve−slope combination sections (exit-to-straight) and other curve−slope combination sections (i.e., approximately 75% of the mean value of others). In contrast, the HDV showed a significant difference in the mean VSP value between the curve−slope combination sections (exit-to-curve) and other curve−slope combination sections (i.e., approximately 67% of the mean value of others).

(4) Across the different alignment units, the mean VSP value for the LDV was higher than that for the HDV, and fluctuations with changes in unit types were greater for the LDV; whereas, HDV exhibited a more stable pattern.

## Correlation between VSP and carbon emissions

MOVES carbon-emission models were established under various operational conditions based on the specified model parameters, yielding the carbon emission equivalents per second for the LDV and HDV under different operational conditions (Table 5). We determined the vehicle's carbon emissions equivalents per second by utilizing the vehicle speed at each second and its VSP. We then summed these values to determine total carbon emissions at each alignment unit, analyzing their correlation with the cumulative sum of the unit's VSP per second.

The results demonstrated a strong linear correlation between total carbon emissions at each alignment unit and the cumulative sum of the VSP per second. Both the LDV and HDV exhibited correlation coefficients of 0.99 (Fig 8).

**Table 5. Carbon emission equivalent value per second.**

| Vehicle type | Fuel type | Vehicle operating conditions | | Emission rate (g/s) |
|---|---|---|---|---|
| | | OpModelID | VSP | |
| HDV | diesel | bin21 | VSP < 0 | 3.491416667 |
| | | bin22 | 0 ≤ VSP < 3 | 9.226972222 |
| | | bin23 | 3 ≤ VSP < 6 | 12.93802778 |
| | | bin24 | 6 ≤ VSP < 9 | 17.45075 |
| | | bin25 | 9 ≤ VSP < 12 | 22.4145 |
| | | bin27 | 12 ≤ VSP < 18 | 29.8975 |
| | | bin28 | 18 ≤ VSP < 24 | 41.33249778 |
| | | bin29 | 24 ≤ VSP < 30 | 52.86861333 |
| | | bin30 | 30 ≤ VSP | 61.23972444 |
| LDV | gasoline | bin33 | VSP < 6 | 2.275566667 |
| | | bin35 | 6 ≤ VSP < 12 | 3.646694444 |
| | | bin37 | 12 ≤ VSP < 18 | 4.749416667 |
| | | bin38 | 18 ≤ VSP < 24 | 6.192777778 |
| | | bin39 | 24 ≤ VSP < 30 | 8.247944444 |
| | | bin40 | 30 ≤ VSP | 10.51344444 |

Based on the aforementioned correlation analysis outcomes, we formulated equations relating total carbon emissions of different alignment units to the cumulative sum of the vehicle VSP per second (Eqs 2–5).

$$m_{CO_{2e}ij} = a \sum VSP_{ij} + b \tag{2}$$

$$m_{CO_{2e}i} = a \sum_{j=1}^{k_i} \sum VSP_{ij} + b \cdot k_i \tag{3}$$

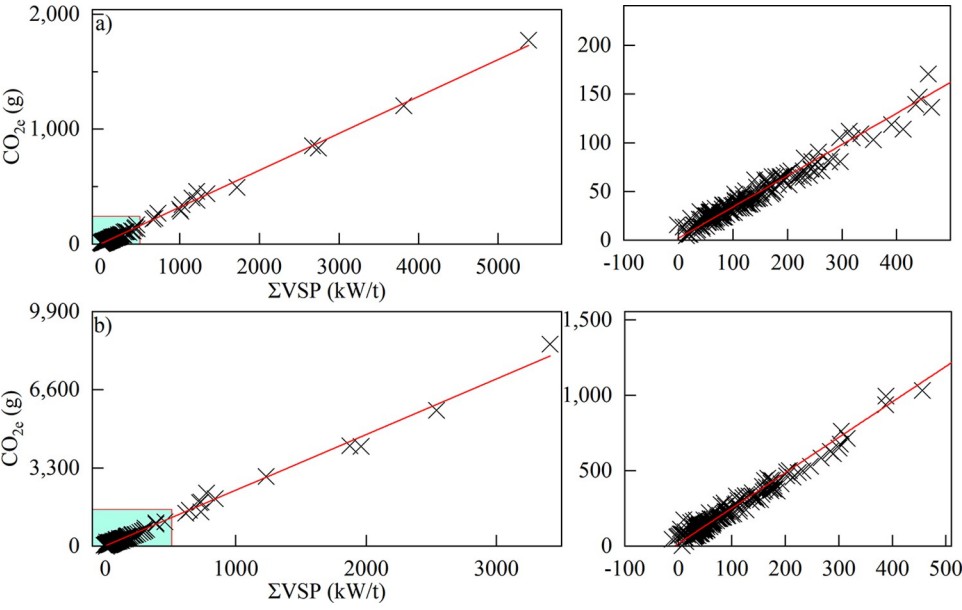

**Fig 8.** Correlation between VSP and carbon emissions from the (a) LDV and (b) HDV.

$$m_{\mathrm{CO_2}ei} = \mathrm{a} \cdot \overline{VSP}_i \cdot S_i + \mathrm{b} \cdot k_i \tag{4}$$

$$m_{\mathrm{CO_2}e\mathrm{total}} = \mathrm{a} \cdot \sum_{i=1}^{n} (\overline{VSP}_i \cdot S_i) + \mathrm{b} \cdot \sum_{i=1}^{n} k_i \tag{5}$$

Here, $m_{\mathrm{CO_2}eij}$ is total carbon emissions of the $j$-th alignment unit of the $i$-th type (g); $\Sigma VSP_{ij}$ is the cumulative sum of VSP per second of the $j$-th alignment unit of the $i$-th type (kW/t); $m_{\mathrm{CO_2}ei}$ is the total carbon emissions of the $i$-th type of alignment unit (g); $k_i$ is the number of the $i$-th type of alignment units; $S_i$ is the vehicle driving time of the $i$-th type of alignment unit (s); $m_{\mathrm{CO_2}e\mathrm{total}}$ is total carbon emissions (g); $n$ is the number of the type of alignment units; $\overline{VSP}_i$ is the mean VSP value of the $i$-th type of alignment unit, (kW/t), and; a and b are constants.

Based on the data of the total carbon emissions and the cumulative sum of the VSP per second of LDV and HDV at each alignment unit, the linear regression analysis method based on the principle of the least squares method was adopted to minimize the sum of squared errors between the regression line and the data points. The following were obtained: the constants a and b for the LDV in Eqs 2–5 were 0.32082 and 2.20158, respectively, and those for the HDV were 2.34937 and 17.58845, respectively. Total carbon emissions along the simulated section of Wenma Expressway were then calculated. Compared with the total carbon emissions derived from the simulation (Table 6), the computed results revealed a minimal discrepancy, suggesting that employing the cumulative sum of the VSP per second across alignment units is an effective method for quantifying highway carbon emissions and assessing carbon emission levels.

In the absence of data from real-vehicle emission tests, the study references the relevant outcomes of prior real-vehicle emission tests to verify the accuracy of the simulation analysis results, as depicted in Table 7.

As indicated in Table 7, the average carbon emissions per kilometer of LDV and HDV in the simulated section of this study are 152 g/km and 748 g/km, respectively, which are close to the existing real-vehicle emission test data. The vehicle operating environment simulated for LDV is similar to that of section 1, both in mountainous plateau areas. However, the simulated value is approximately 7% lower than the measured value of section 1; this might be because the altitude range of section 1 is relatively higher than that of the simulated section, the section type includes both ordinary highways and expressways, and the maximum grade of section 1 reaches 20%, making the vehicle load and the influence of the plateau environment more prominent. Sections 2 and 3 have relatively flat terrain and good alignment. The carbon emission factors are reduced by about 10% and 40%, respectively, compared with the simulated section. The altitude difference between sections 2 and 3 is close to 1000 m. The reduction in air-fuel ratio and the attenuation of vehicle power performance in the high-altitude, low-pressure, and low-oxygen environment may be the main factors leading to the increase in vehicle carbon emission factors.

The simulated value of HDV is slightly higher than the measured value of section 5 and is at the upper limit of the measured carbon emission range. Section 5 is located in the Guanzhong Plain, which has flat terrain and good alignment. In contrast, the simulated section is in a mountainous plateau area. The low-oxygen environment on the plateau, the frequent changes

**Table 6. Total carbon emissions.**

| Vehicle type | LDV | | HDV | |
|---|---|---|---|---|
| | Simulated value | Calculated value | Simulated value | Calculated value |
| Total carbon emissions (g) | 18060 | 18062 | 89054 | 89139 |

**Table 7. Relevant real-vehicle emission test data.**

| Data source | References (real-vehicle emission tests) | | | | | Simulation section of this study | |
|---|---|---|---|---|---|---|---|
| | [17] | [30] | [30] | [31] | [18] | | |
| Test area | Xining–Maduo | Kunming | Kaiyuan | Xining | Xi'an–Baoji | Wenchuan, Lixian, Barkam | |
| Section type | section 1: expressway, ordinary highway | section 2: expressway | section 3: expressway | Section 4: urban area, suburban area and expressway | section 5: expressway | expressway | |
| Terrain | steep | relatively flat | relatively flat | relatively flat | flat | steep | |
| Altitude range (m) | 2000–3000 | 1960 | 1110 | 2350 | 500–700 | 1300–3000 | 1300–3000 |
| Vehicle type | LDV | LDV | LDV | HDV | HDV | LDV | HDV |
| | passenger car | passenger car | passenger car | cargo truck | cargo truck (four-axle) | passenger car | cargo truck (four-axle) |
| Fuel types | gasoline | gasoline | gasoline | diesel | diesel | gasoline | diesel |
| Carbon emission factors (g/km) | 162–165 | 137 | 90 | 428 | 550–750 | 152 | 748 |

in vehicle speed, and the increase in vehicle load are all important factors contributing to the increase in vehicle carbon emissions.

The above comparative analysis with the real-vehicle emission test data illustrates the accuracy of the coupled simulation of CarSim/TruckSim and MOVES adopted in this study for analyzing vehicle carbon emissions. In addition, the carbon emission factors of passenger cars and cargo trucks in the greenhouse gas emission inventory released by the Ministry of Ecology and Environment of China are 174.9 g/km and 965.6 g/km respectively. The carbon emission factor of LDV in the inventory is close to the measured and simulated values. At the same time, there is a relatively large difference for HDV, with an increase of approximately 30%. The HDV in the inventory model covers cargo truck types such as four-axle, five-axle, and six-axle, as well as different levels of road scenarios, with a wider scope involved. Moreover, the estimation of activity level data may be rather conservative, resulting in relatively higher carbon emission factors.

The real-vehicle emission test data of sections 2 and 4 were selected to verify the validity of the relevant Eq 5. The test section 2 is 17.91 km, including parts of the G56 Hangrui Expressway and the Kunming East Ring Expressway. The cumulative sum of the VSP of LDV per second is 7942 kW/t, and the number of various alignment units is approximately 38. The total carbon emissions calculated according to Eq 5 is 2632 g, approximately 7.6% higher than the measured value of 2445 g. The test section 4 is 141.7 km, and the cumulative sum of the VSP of HDV per second is 20227 kW/t. The test data lacks alignment indicators. Referring to the average of 2.1 alignment units per kilometer of expressways in mountainous plateau areas in this study, considering that the alignment indicators of ordinary roads are relatively low, the number of various alignment units of section 4 is calculated as 375. The total carbon emissions obtained are 54116 g, approximately 10.3% lower than the measured value of 60305 g. The considerable difference in the calculation results may be related to the type of the test section, which includes urban and suburban roads and expressways. The operating conditions of vehicles in urban sections are significantly different from those on expressways, which has a greater impact on the carbon emissions of HDV. The verification and analysis using the real-vehicle emission test data of sections 2 and 4 illustrate the validity of the relevant Eq 5.

## Influencing factors of vehicle carbon emissions

### Road alignment conditions

In China and abroad, when calculating the carbon emissions of road vehicles, carbon emission factors or coefficients are typically based on carbon emissions per kilometer, and the

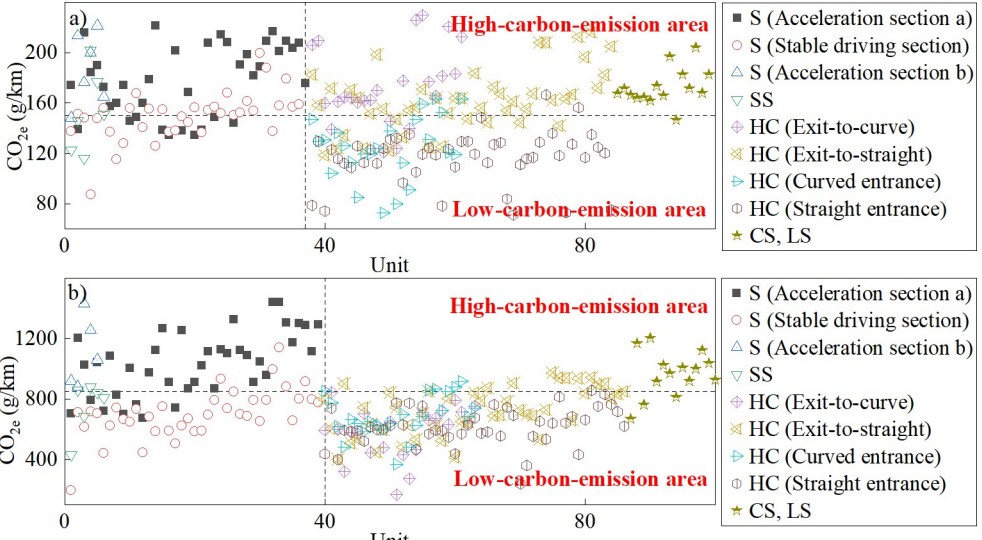

**Fig 9.** Carbon emission per kilometer at each alignment unit of the (a) LDV and (b) HDV.

calculation of carbon emissions per kilometer is often carried out under fixed spacing, time periods or operating conditions. In this study, by calculating the average carbon emission equivalent per kilometer for each alignment unit, we analyzed the distribution of carbon emission equivalents per kilometer for different alignment unit types. The statistical results are depicted in Fig 9. Here, the "S (acceleration section a)" and "S (stable driving section)" denote the two phases (i.e., acceleration driving towards the expected speed, and stable driving or deceleration) on the same straight section of the road. The "S (acceleration section b)" signifies the section where the vehicle speed had not reached the expected speed.

Distribution intervals of the carbon emission equivalent per kilometer for the LDV and HDV at each alignment unit were 60–240 g and 100–1600 g, respectively. Using the median values (150 g and 850 g, respectively) within these intervals as threshold values, the carbon emission equivalents of LDV and HDV were equally categorized into high- and low-carbon-emission areas, respectively. The findings indicated the following:

(1) In straight sections, when vehicles were driving stably or decelerating, the carbon emission equivalent per kilometer for the LDV was evenly distributed around the threshold value, whereas that for the HDV was within the low-carbon-emission area. Conversely, during acceleration, both the LDV and HDV exhibited carbon emission equivalents in the high-carbon-emission area.

(2) In the curve–slope combination and longitudinal slope sections, the carbon emission equivalent per kilometer for both the LDV and HDV was in the high-carbon-emission area.

(3) In the horizontal curve sections, the carbon emission equivalent per kilometer for the LDV tended to be higher in the exit sections than in the entrance sections, placing the former in the high-carbon-emission area and the latter in the low-carbon-emission area. For the HDV, the difference in carbon emission equivalent per kilometer between the exit and entrance sections was relatively minor, and both were within the low-carbon-emission area.

(4) In the Wenchuan to Lixian section (K0–K64), the average carbon emission equivalents per kilometer for LDV and HDV are 144 g and 689 g, respectively. In the Lixian to Barkam section (K64–K119), the average carbon emission equivalents per kilometer for LDV and HDV are 160 g and 817 g, respectively. The main reason for the increase in the average carbon

**Table 8. Correlation degree between carbon emissions per kilometer and various factors.**

| Vehicle type | Factors | S | | | | HC | | | | CS, LS |
|---|---|---|---|---|---|---|---|---|---|---|
| | | Acceleration section a | Stable driving section | Acceleration section b | SS | Exit-to-curve | Exit-to-straight | Curved entrance | straight entrance | |
| LDV | R | — | — | — | — | 0.7 | 0.73 | 0.71 | 0.74 | — |
| | t | 0.59 | 0.64 | 0.63 | 0.53 | 0.58 | 0.56 | 0.63 | 0.62 | 0.59 |
| HDV | R | — | — | — | — | 0.66 | 0.68 | 0.67 | 0.72 | — |
| | t | 0.60 | 0.68 | 0.48 | 0.48 | 0.64 | 0.59 | 0.59 | 0.63 | 0.69 |

In the grey relational analysis, the influence factor R was converted to curvature (1/R).

emission equivalents per kilometer for both LDV and HDV in the latter section compared to the former is that the increase in the average grade leads to a rise in the VSP. The mean VSP value has increased by 1.18 times and 1.33 times, respectively, while the average carbon emission equivalents per kilometer have increased by 1.11 times and 1.19 times, respectively. The change in the average grade has a more significant impact on HDV.

The grey relational analysis method was employed to investigate the correlation between the carbon emissions per kilometer at each alignment unit and the average grade of the unit, as well as the circular curve radius. It can capture the dynamic changes in vehicle carbon emissions of alignment units with different combinations of grade and radius and the dynamic evolution of these emissions with road alignment conditions. It can also analyze the changing trend of the correlation degree between vehicle carbon emissions and road alignment conditions, and explore the dynamic connection between them. A comparative analysis of the correlations among these factors was conducted by quantifying the correlation degree between unit carbon emissions per kilometer and various factors (Table 8).

The results showed that in the horizontal curve section, the correlation degree between the carbon emissions equivalent per kilometer of the LDV and HDV and the circular curve radius was higher than that with the average grade, indicating that radius had a greater influence than average grade on the carbon emissions equivalent per kilometer. In the straight section, when the LDV and HDV were driving stably or decelerating at their expected speeds, the correlation degree between the average grade of the unit and the carbon emission equivalent per kilometer was higher than that under other acceleration operating conditions. In the curve–slope combination and longitudinal slope sections, the correlation degree between the carbon emission equivalent per kilometer of the HDV and the average grade of the unit was greater than that of the LDV. This was primarily because the tendency of the HDV speed to decrease when moving upslope was more pronounced than the LDV speed.

## Environmental characteristics

Altitude, temperature, and humidity significantly affect carbon emissions from expressways in mountainous plateau areas. Climatic characteristics of these areas exhibit significant vertical differences with altitude changes. Specifically, the temperature decreases with increasing altitude, and the vertical temperature lapse rate is greater in summer than in winter. Furthermore, water vapor pressure decreases with increasing altitude. During summer, relative humidity in the upper area of the mountain is higher than in the lower area; the opposite is true during winter.

Based on the monthly average temperature and humidity data at different altitudes of the Wenma Expressway (Fig 10) and utilizing the operating speed prediction model for the Wenma Expressway to establish the average speed distribution conditions (Table 9), a

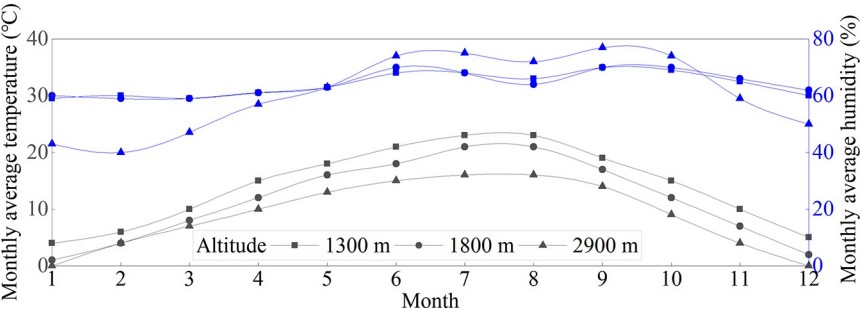

**Fig 10. Monthly average temperature and humidity at different altitudes.** Data are the average monthly temperature and humidity values for each month in recent 10 years and were obtained from the China Weather website, http://www.weather.com.cn/.

**Table 9. Average speed distribution conditions of MOVES county scale model.**

| LDV | | HDV | |
|---|---|---|---|
| Speed range (km/h) | Proportion (%) | Speed range (km/h) | Proportion (%) |
| 76 ≤ Speed < 85 | 0.2006 | 60 ≤ Speed < 68 | 9.1365 |
| 85 ≤ Speed < 93 | 0.6267 | 68 ≤ Speed < 76 | 17.6997 |
| 93 ≤ Speed < 101 | 5.9915 | 76 ≤ Speed < 85 | 73.1638 |
| 101 ≤ Speed < 109 | 36.2246 | | |
| 109 ≤ Speed < 117 | 56.9566 | | |

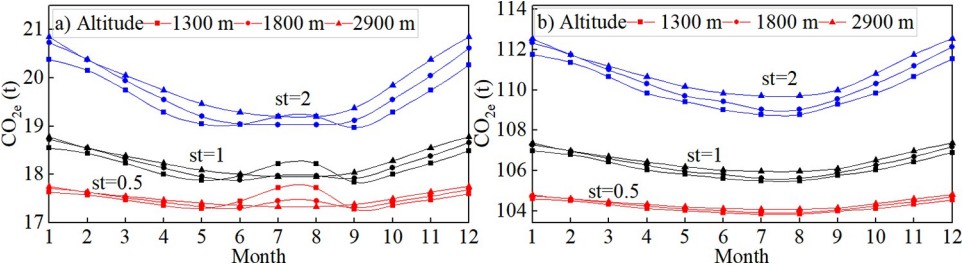

**Fig 11.** Monthly carbon emission change curve at different altitudes of the (a) LDV and (b) HDV. st, average number of vehicle starts.

mesoscopic carbon emission model in MOVES was developed under various climatic conditions. It has been proposed that there are 1000 LDVs and 1000 HDVs, respectively. Each vehicle travels a distance of 100 km, and the total mileage of both types of vehicles is 100000 km. Carbon emission equivalents during vehicle start-up and operation were calculated.

Monthly change trends of the carbon emission equivalent during vehicle start-up and operation at altitudes of 1300, 1800, and 2900 m are illustrated in Fig 11.

The results showed the following:

(1) Carbon emissions from the HDV were significantly higher in winter than in summer, with carbon emissions displaying a concave distribution. For the LDV, the carbon emission trend exhibited a concave distribution only at an altitude of 2900 m; whereas, at altitudes of 1300 m and 1800 m, vehicle carbon emissions demonstrated a "hump-shaped" distribution. Variations in climatic conditions can significantly affect vehicle carbon emissions. At excessively high temperatures, engines may overheat or experience a vapor lock, negatively affecting

engine performance and leading to increased fuel consumption and carbon emissions. Conversely, engine start-up becomes difficult at excessively low temperatures, heat dissipation losses increase, and engine power decreases, resulting in higher fuel consumption and carbon emissions.

(2) Vehicle carbon emissions were closely correlated with the number of starts. The more frequently the vehicle was started, the greater the carbon emissions, with the differences being the most pronounced during winter. In winter, when the LDV was started on an average of two times, the carbon emission equivalent was approximately 1.17-times higher than when the LDV was started on an average of 0.5 times; whereas, for the HDV, this ratio was 1.07. In summer, when the LDV was started on an average of two times, the carbon emission equivalent was approximately 1.09-times higher than when the LDV was started on an average of 0.5 times; for the HDV, this ratio was 1.04. When a vehicle starts, the engine has to overcome internal friction and static inertia. Inefficient fuel injection atomization and uneven air-fuel mixing lead to incomplete combustion, increasing the vehicle's carbon emissions. In winter, low temperatures result in poor fuel atomization, increased oil viscosity, and greater engine frictional resistance. In contrast, during summer, the fuel atomization is relatively better, the oil viscosity is lower, and the engine friction is smaller. Consequently, frequent vehicle starts in summer cause a smaller increase in carbon emissions than in winter.

## Conclusions

The main conclusions of this study are as follows:

(1) The change trends of VSP for the LDV and HDV in the Wenma Expressway were similar, with both distributions closely resembling a normal distribution. The average grade of the Lixian–Barkam (K64–K119) section increased from 1% to 1.76% compared with that of the Wenchuan–Lixian (K0–K64) section. The mean VSP value of the LDV increased from 12.57 kW/t to 14.79 kW/t, and that of the HDV increased from 5.22 kW/t to 6.93 kW/t.

(2) VSP distributions for the LDV and HDV in the horizontal curve and straight sections were relatively broad. In contrast, VSP distributions in the curve–slope combination and longitudinal slope sections were more concentrated. Furthermore, the VSP distribution in the short straight section was more uniform than that in other sections. Additionally, the mean VSP value distributions for the alignment units in the HC (exit-to-curve), HC (exit-to-straight), HC (curved entrance), HC (straight entrance), and straight sections followed a normal distribution, with significance > 0.05.

(3) There was a strong correlation between total carbon emissions at each alignment unit and the cumulative sum of the VSP per second of the unit. Correlation coefficients for both the LDV and HDV reached 0.99, supporting their use in calculating highway carbon emissions and evaluating carbon emission levels. Moreover, the distribution intervals of the carbon emission equivalent per kilometer for the LDV and HDV at each alignment unit were 60–240 g and 100–1600 g, respectively. In the expressway design phase, the variation trends of vehicle operating speeds for diverse alignment design alternatives are computed and analyzed. Employing the calculation model integrating the total carbon emissions of alignment units and the cumulative sum of VSP per second, the total carbon emissions of both LDV and HDV are acquired, facilitating an effective evaluation of the carbon emission levels of expressway alignment design schemes in mountainous plateau areas.

(4) Both the circular curve radius and average grade of the unit significantly influenced carbon emissions per kilometer. In the horizontal curve section, the effect of the radius on carbon emissions was greater than that of the average grade; this should attract further attention in the future for the optimization of the horizontal alignment design of expressways in mountainous

plateau areas, especially for the combination design of horizontal alignment, which determines the operating speed of vehicles. In straight sections, the average grade correlated more strongly with carbon emissions per kilometer while driving stably or decelerating than in acceleration operating conditions. In the curve−slope combination and longitudinal slope sections, the correlation degree between carbon emissions per kilometer and the average grade was stronger for the HDV than for the LDV.

(5) Carbon emissions from the HDV were significantly higher during winter than during summer, exhibiting a concave distribution trend. In contrast, carbon emission trends for the LDV displayed a "hump-shaped" distribution at altitudes of 1300 m and 1800 m, and a concave distribution at an altitude of 2900 m. Moreover, frequent vehicle starts had a substantial impact on carbon emissions, with a notably greater effect in winter than in summer. When the average number of starts for the LDV and HDV was two during summer, the corresponding carbon emission equivalents were approximately 1.09- and 1.04-times higher than those when the average number of starts was 0.5, respectively; in winter, these factors increased by 1.17- and 1.07-times, respectively.

## Supporting information

**S1 Table. Carbon emission equivalent per kilometer at each alignment unit of the (a) LDV and (b) HDV.**
(DOCX)

## Acknowledgments

We are grateful to the Sichuan Highway Planning, Survey, Design and Research lnstitute ltd. for generously providing the highway design documents of the test routes.

## Author Contributions

**Conceptualization:** Jianping Gao, Yunyong He.

**Data curation:** Changfeng Yang.

**Formal analysis:** Xin Huang.

**Funding acquisition:** Jianping Gao, Yunyong He.

**Investigation:** Enhuai He.

**Methodology:** Jianping Gao, Xin Huang.

**Project administration:** Jianping Gao, Yunyong He.

**Resources:** Lu Sun.

**Software:** Xin Huang.

**Supervision:** Jianping Gao, Yunyong He.

**Validation:** Xin Huang.

**Visualization:** Xin Huang.

**Writing – original draft:** Xin Huang.

**Writing – review & editing:** Jianping Gao, Xin Huang.

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
