## [Decision Letter · Decision Letter 0]

1 Dec 2024

PONE-D-24-43843Analysis of vehicle carbon emission characteristics on expressways in mountainous plateau areas based on the coupled simulation of CarSim/TruckSim and MOVESPLOS ONE

Dear Dr. HE,

Thank you for submitting your manuscript to PLOS ONE. After careful consideration, we feel that it has merit but does not fully meet PLOS ONE’s publication criteria as it currently stands. Therefore, we invite you to submit a revised version of the manuscript that addresses the points raised during the review process.

We look forward to receiving your revised manuscript.

Kind regards,

Zhihong (Arry) Yao, Ph.D.

Academic Editor

PLOS ONE

Reviewers' comments:

Reviewer's Responses to Questions

**Comments to the Author**

1. Is the manuscript technically sound, and do the data support the conclusions?

Reviewer #1: Partly

Reviewer #2: Yes

Reviewer #3: Partly

2. Has the statistical analysis been performed appropriately and rigorously? 

Reviewer #1: No

Reviewer #2: Yes

Reviewer #3: Yes

3. Have the authors made all data underlying the findings in their manuscript fully available?

Reviewer #1: Yes

Reviewer #2: Yes

Reviewer #3: No

4. Is the manuscript presented in an intelligible fashion and written in standard English?

Reviewer #1: Yes

Reviewer #2: Yes

Reviewer #3: Yes

5. Review Comments to the Author

Reviewer #1: This paper employed the coupling simulation analysis method combining CarSim/ TruckSim and MOVES to study the carbon emission characteristics of expressways in the western Sichuan Plateau mountainous. Overall, the numerical method and obtained results are traditional. It is advised to emphasize the key novel findings in the introduction and conclusions. And more, I hope that the following review comments can help the authors to improve the quality of this paper:

1. How does harsh climatic characteristics in plateau affect the release of carbon emissions, and how to consider these in this paper?

2. In the section of dynamic simulation: What are the actual stress states and operating conditions in this study?

3. From Fig. 1, some parameters can be obtained. However, the name of Fig.1 is road model. How to get the model from it?

4. Some tables are confusing, and some auxiliary lines should not be deleted.

5. In page 17, is it the 1000000km?

6. In the tables, the proportion should be percentage.

7. In the equations, some expressions are not standardized and should be changed, such as mco2eij and HC(..).

8. What are the characteristics of section K64‒K119 and section K0‒K64?

Reviewer #2: 1. In this paper, Carsim/Trucksim and MOVES are combined to simulate different vehicle models on expressway, and the data are analyzed, then the VSP and emission characteristics of vehicles under different road alignment and environmental conditions are obtained, and the analysis conclusions are obtained. However, it is suggested to highlight the innovative points of the paper.

2. In this paper authors applied two car models in simulation, whether these two models are representative in highway scene, whether they are the main proportion of highway traffic flow, it is suggested to add some content to show that.

3. The conclusion of this paper mainly comes from simulation data, so the accuracy of simulation is important, which needs to be supplemented.

Reviewer #3: This paper examines the carbon emission characteristics of mountain highways based on CarSim/TruckSim and MOVES. The document is well-organized and easy to follow. However, I have a few concerns about the previous version.

1.In the section titled "Road Alignment Conditions," grey relational analysis was utilized to examine the relationship between carbon emissions and other parameters. Please provide the reasons for choosing this method. Is GRA widely used in analyzing factors related to emissions?

2.Please provide a detailed process for calibrating parameters a and b in Equations (2), (3), (4), and (5) rather than simply stating "Based on the statistical analysis outcomes."

3.In the analysis of the results presented in Figure 11, a brief explanation is given for the reasons behind the differences in carbon emissions caused by temperature in result (1). A similar explanation should be provided in result (2) regarding the causes of carbon emission differences due to the vehicle starting frequency.

4.This paper analyzes the vehicle carbon emission characteristics based on the simulated data. Is there any corresponding real-world data to support the model presented in this paper?

5.The mountainous plateau region studied in this paper is a unique terrain. Comparing and discussing the vehicle carbon emission characteristics in this area with those of expressways in plain regions reported in other studies would provide valuable insights.

6. PLOS authors have the option to publish the peer review history of their article (what does this mean?). If published, this will include your full peer review and any attached files.

Reviewer #1: No

Reviewer #2: No

Reviewer #3: No

---

## [Author Response · Author response to Decision Letter 0]

13 Jan 2025

Reviewer 1

Comment 1: How does harsh climatic characteristics in plateau affect the release of carbon emissions, and how to consider these in this paper?

• Response:

o We sincerely appreciate the valuable comments. Subsequently, we have supplemented the influence process of the harsh climate characteristics of the plateau on vehicle carbon emissions, illustrating that in a low-pressure and low-oxygen environment, the intake volume of the engine is significantly reduced, resulting in incomplete combustion. In a low-temperature environment, the engine dissipates heat quickly, the fuel atomization is poor, and the combustion efficiency is low. All these effects will lead to an increase in vehicle fuel consumption and carbon emissions. The revised content is presented in "Introduction", Paragraph 4, from Line 81 to Line 85. This study on the effect of the harsh climate characteristics of the plateau on vehicle carbon emissions focuses on analyzing the influence law of the difference characteristics of temperature and humidity in mountainous plateau areas with changes in altitude and season on vehicle carbon emissions. The revised content is presented in "Introduction", Paragraph 4, from Line 106 to Line 109. We constructed a localized MOVES model and selected Denver, USA, as the region in the model, which is similar to the plateau scenario simulated in this study. Subsequently, we consulted the temperature and humidity data of different altitudes (1300 m, 1800 m, and 2900 m) in different seasons in the simulated region over the past decade and focused on simulating and analyzing the variation characteristics of vehicle carbon emissions on mountainous plateau areas expressways under different altitude and season scenarios. These modifications have enhanced the clarity and readability of the manuscript.

Comment 2: In the section of dynamic simulation: What are the actual stress states and operating conditions in this study?

• Response:

o We sincerely appreciate the valuable comments. Subsequently, we have supplemented the description of the actual stress states and operating conditions in the vehicle dynamic simulation of this study. The model takes into account the driving force, braking force, rolling resistance, air resistance, tire lateral force, and inertial force that the vehicle experiences during the driving process. Moreover, based on formulating the target speed curve, it simulates and analyzes the operating conditions of the vehicle such as the acceleration, speed and VSP on a second-by-second basis. The revised content is presented in "Dynamic simulation", Paragraph 1, from Line 115 to Line 119. The target speed curve was derived from the operating speed prediction model in the "Specification for Highway Safety Audit (JTJ B05-2015)", which represents the upper speed limit at which 85% of vehicles can travel naturally. The vehicle operating speed was calculated by setting parameters such as the expected speed, minimum speed, and acceleration to increase the universality and representativeness of the simulation test. The revised content is presented in "Operating speed prediction", Paragraph 1, from Line 152 to Line 154. We have corrected the relevant figure. The original figure "Fig 3. Predicted vehicle operating speeds" has been revised to "Fig 3. Target curves and actual value of the vehicle operating speed", which can be found in "Operating speed prediction", Paragraph 2, Line 165. Besides, we added the scatter points of the vehicle's speed per second obtained from the vehicle dynamic simulation to the figure. These modifications have enhanced the clarity and readability of the manuscript.

Comment 3: From Fig. 1, some parameters can be obtained. However, the name of Fig.1 is road model. How to get the model from it?

• Response:

o We sincerely appreciate the valuable comments. Subsequently, the original figure "Fig 1. Road model" has been revised to "Fig 1. Road geometry", which includes road alignment and superelevation and can be found in "Road model", Paragraph 1, Line 133. Besides, we have supplemented and explained that the road has two-way four lanes, and the pavement type is asphalt concrete. These contents are essential components for the construction of the road model. The revised content is presented in "Road model", Paragraph 1, from Line 130 to Line 131. These corrections have enhanced the clarity and readability of the manuscript.

Comment 4: Some tables are confusing, and some auxiliary lines should not be deleted.

• Response:

o We were really sorry for our careless mistakes. Subsequently, we have added auxiliary lines to the tables in accordance with the requirements for tables in the manuscript body formatting guidelines of PLOS. The revised locations involve all the tables in the manuscript. These corrections have enhanced the clarity and readability of the manuscript.

Comment 5: In page 17, is it the 1000000km?

• Response:

o We appreciate the reviewer's astute observation. We verified that the "100000 km" at this position in the original text was correct. The original text "It has been proposed that there are 1000 LDVs and 1000 HDVs with a total mileage of 100000 km for each vehicle type" has been revised to "It has been proposed that there are 1000 LDVs and 1000 HDVs, respectively. Each vehicle travels a distance of 100 km, and the total mileage of both types of vehicles is 100000 km", which can be found in "Environmental characteristics", Paragraph 2, from Line 394 to Line 396. This modification has enhanced the clarity and readability of the manuscript.

Comment 6: In the tables, the proportion should be percentage.

• Response:

o We appreciate the reviewer's astute observation. We have corrected the relevant table. The content in original table 8 "0.002006, 0.091365, 0.006267, etc." has been revised to "0.2006%, 9.1365%, 0.6267%, etc.", which can be found in "Environmental characteristics", Paragraph 2, table 9, Line 400. These corrections have enhanced the clarity and readability of the manuscript.

Comment 7: In the equations, some expressions are not standardized and should be changed, such as mco2eij and HC(..).

• Response:

o We were really sorry for our careless mistakes. We have corrected the relevant Equations. The content in original Equations 2–5 " , , " has been revised to " , , ", which can be found in "Correlation between VSP and carbon emissions", Paragraph 3, Equations 2–5, from Line 264 to Line 267. Besides, we have corrected the relevant text. The original text "HC (exit-to-curve; 11.7 kW/t); HC (straight entrance; 3.9 kW/t)" has been revised to "HC (exit-to-curve), at 11.7 kW/t; HC (straight entrance), at 3.9 kW/t ", which can be found in "VSP distribution in different alignment units", Paragraph 6, from Line 233 to Line 234. These corrections have enhanced the clarity and readability of the manuscript.

Comment 8: What are the characteristics of section K64‒K119 and section K0‒K64?

• Response:

o We think this is an excellent suggestion. Subsequently, we have supplemented the description of the characteristics of section K64‒K119 and section K0‒K64. On the one hand, we have supplemented the characteristics of the topography and landforms of the simulated road sections. The Wenma Expressway is located in the interlaced contact zone between the Qinghai-Tibet Plateau's eastern margin and the Sichuan Basin's northwestern edge. Among them, the Wenchuan to Lixian section (K0‒K64) is dominated by medium and low mountains and river valley terraces, with relatively wide valleys. The Lixian to Barkam section (K64‒K119) mainly features alpine and gorge landforms, with steep mountains and deep valleys. The revised content is presented in "Road model", Paragraph 1, from Line 127 to Line 130. This modification supplemented and explained the construction conditions of the simulated road section. On the other hand, the characteristics of vehicle carbon emissions in the former and latter road sections were supplemented and analyzed. The mean VSP value of LDV and HDV of the section K64‒K119 has increased by 1.18 times and 1.33 times than that of the section K0‒K64 respectively, while the average carbon emission equivalents per kilometer have increased by 1.11 times and 1.19 times, respectively. The revised content is presented in "Road alignment conditions", Paragraph 6, from Line 356 to Line 363. This modification have enhanced the clarity and readability of the manuscript.

Reviewer 2

Comment 1: In this paper, Carsim/Trucksim and MOVES are combined to simulate different vehicle models on expressway, and the data are analyzed, then the VSP and emission characteristics of vehicles under different road alignment and environmental conditions are obtained, and the analysis conclusions are obtained. However, it is suggested to highlight the innovative points of the paper.

• Response:

o We think this is an excellent suggestion. Subsequently, we have emphasized the key novel findings in the introduction and conclusions. we have supplemented the contributions of our research to the literature on the design, planning and management of expressways in mountainous plateau areas. Especially the carbon emission calculation model based on the cumulative sum of VSP of alignment units constructed by us can obtain the carbon emissions of different alignment design schemes during the design stage of expressways in mountainous plateau areas, and achieve an effective evaluation of the carbon emission levels of expressway alignment design schemes in mountainous plateau areas, providing a fundamental support for low-carbon alignment design in these areas. In addition, in the existing studies, the vehicle carbon emission factors were mostly based on fixed spacing, time periods or operating conditions. This study analyzed the distribution characteristics of vehicle carbon emission factors in different alignment units of expressways in mountainous plateau areas and quantified the effect degree of key alignment indicators on vehicle carbon emissions. The revised content is presented in "Introduction", Paragraph 4, from Line 90 to Line 109, and in "Conclusions", specifically in Paragraph 4 from Line 443 to Line 448 and in Paragraph 5 from Line 451 to Line 453. These modifications have enhanced the clarity and readability of the manuscript.

Comment 2: In this paper authors applied two car models in simulation, whether these two models are representative in highway scene, whether they are the main proportion of highway traffic flow, it is suggested to add some content to show that.

• Response:

o We sincerely appreciate the valuable comments. Based on the traffic composition data of the Wenma Expressway in the past four years, we have supplemented the proportions of the two types of vehicle models selected in this study among LDV and HDV respectively, in order to illustrate the representativeness of the two vehicle models in the highway scenarios. The revised content is presented in "Vehicle model", Paragraph 1, from Line 137 to Line 142. This modification have enhanced the clarity and readability of the manuscript.

Comment 3: The conclusion of this paper mainly comes from simulation data, so the accuracy of simulation is important, which needs to be supplemented.

• Response:

o We think this is an excellent suggestion. In the absence of data from real-vehicle emission tests, we have referenced the relevant outcomes of prior real-vehicle emission tests to verify the accuracy of the simulation analysis results. We have supplemented the sources of relevant outcomes of prior real-vehicle emission tests data and integrated the areas of the test sections, section types, terrains, altitude ranges, vehicle types, fuel types and the carbon emission factors obtained from the tests, which can be found in "Correlation between VSP and carbon emissions", Paragraph 5, table 7, Line 287. We selected two test sections, namely Section 1 and Section 5, where the test vehicle and fuel types were similar to those in the simulation tests of this study. Real-vehicle emission tests for LDV and HDV were carried out on these two sections respectively. Subsequently, we compared the differences in carbon emission factors obtained from the simulation tests and the real-vehicle emission tests, and analyzed the possible reasons for the differences from aspects such as the types, terrains and altitude ranges of the test sections. The average carbon emissions per kilometer of LDV and HDV in the simulated section of this study are 152 g/km and 748 g/km, respectively, which are close to the existing real-vehicle emission test data. The comparative analysis with the real-vehicle emission test data illustrates the accuracy of the coupled simulation of CarSim/TruckSim and MOVES adopted in this study for analyzing vehicle carbon emissions. The revised content is presented in "Correlation between VSP and carbon emissions", Paragraph 6‒8, from Line 288 to Line 307. In addition, we also conducted a comparative analysis of the carbon emission factors of LDV and HDV on the simulated road sections in this study and the data in the greenhouse gas emission inventory released by the Ministry of Ecology and Environment of China. The revised content is presented in "Correlation between VSP and carbon emissions", Paragraph 8, from Line 307 to Line 313. These modifications have strengthened the scientific validity and reliability of our study, ensuring that our conclusions are more robust and well-founded.

Reviewer 3

Comment 1: In the section titled "Road Alignment Conditions," grey relational analysis was utilized to examine the relationship between carbon emissions and other parameters. Please provide the reasons for choosing this method. Is GRA widely used in analyzing factors related to emissions?

• Response:

o We sincerely appreciate the valuable comments. Subsequently, we have supplemented the reasons for choosing grey relational analysis in this study. This method can capture the dynamic changes in vehicle carbon emissions of alignment units with different combinations of grade and radius and the dynamic evolution of these emissions with road alignment conditions. It can also analyze the changing trend of the correlation degree between vehicle carbon emissions and road alignment conditions, and explore the dynamic connection between them. The revised content is presented in " Road alignment conditions", Paragraph 7, from Line 366 to Line 369. This modification have enhanced the clarity and readability of the manuscript.

o Grey relational analysis has been used to analyze the influencing factors of vehicle exhaust pollutant emissions. For example, in the literature "Fu MQ. Comprehensive analysis of heavy-duty diesel vehicle emissions under the influence of multiple factors. Tianjin University of Technology, 2024", the grey relational analysis was adopted to analyze the pollutant emission characteristics of heavy-duty diesel vehicles under the influence of multiple factors.

Comment 2: Please provide a detailed process for calibrating parameters a and b in Equations (2), (3), (4), and (5) rather than simply stating "Based on the statistical analysis outcomes.

• Response:

o We appreciate the reviewer's astute observation. Subsequently, we have supplemented a detailed process for calibrating parameters a and b in Equations 2‒5. Based on the data of the total carbon emissions and the cumulative sum of the VSP per second of LDV and HDV at each alignment unit, the linear regression analysis method based on the principle of the least squares method was adopted to minimize the sum of squared errors between the regression line and the data points. The following were obtained: the constants a and b for the LDV in Equations 2‒5 were 0.32082 and 2.20158, respectively, and those for the HDV were 2.34937 and 17.58845, respectively. The revised content is presented in "Correlation between VSP and carbon emissions", Paragraph 4, from Line 274 to Line 277. This modification have enhanced the clarity and readability of the manuscript.

Comment 3: In the analysis of the results presented in Figure 11, a brief explanation is given for the reasons behind the differences in carbon emissions caused by temperature in result (1). A similar explanation should be provided

---

## [Decision Letter · Decision Letter 1]

21 Jan 2025

Analysis of vehicle carbon emission characteristics on expressways in mountainous plateau areas based on the coupled simulation of CarSim/TruckSim and MOVES

PONE-D-24-43843R1

Dear Dr. HE,

We’re pleased to inform you that your manuscript has been judged scientifically suitable for publication and will be formally accepted for publication once it meets all outstanding technical requirements.

Kind regards,

Zhihong (Arry) Yao, Ph.D.

Academic Editor

PLOS ONE

Additional Editor Comments (optional):

Reviewers' comments:

Reviewer's Responses to Questions

**Comments to the Author**

1. If the authors have adequately addressed your comments raised in a previous round of review and you feel that this manuscript is now acceptable for publication, you may indicate that here to bypass the “Comments to the Author” section, enter your conflict of interest statement in the “Confidential to Editor” section, and submit your "Accept" recommendation.

Reviewer #1: All comments have been addressed

Reviewer #2: All comments have been addressed

Reviewer #3: All comments have been addressed

2. Is the manuscript technically sound, and do the data support the conclusions?

Reviewer #1: Yes

Reviewer #2: Yes

Reviewer #3: Yes

3. Has the statistical analysis been performed appropriately and rigorously? 

Reviewer #1: Yes

Reviewer #2: Yes

Reviewer #3: Yes

4. Have the authors made all data underlying the findings in their manuscript fully available?

Reviewer #1: Yes

Reviewer #2: Yes

Reviewer #3: No

5. Is the manuscript presented in an intelligible fashion and written in standard English?

Reviewer #1: Yes

Reviewer #2: Yes

Reviewer #3: Yes

6. Review Comments to the Author

Reviewer #1: 1.The significant digits should be considered, such as the Table 5.

2. The units should be corrected accrodig to standards, such as t.

3. English should be improved including article, sigular or plural of nouns.

Reviewer #2: I have no more comments for authors, their answers meet the reviews' comments and their works have been done well.

Reviewer #3: (No Response)

7. PLOS authors have the option to publish the peer review history of their article (what does this mean?). If published, this will include your full peer review and any attached files.

Reviewer #1: No

Reviewer #2: No

Reviewer #3: No

---

## [Editor Report · Acceptance letter]

27 Jan 2025

PONE-D-24-43843R1 

PLOS ONE

Dear Dr. HE, 

I'm pleased to inform you that your manuscript has been deemed suitable for publication in PLOS ONE. Congratulations! Your manuscript is now being handed over to our production team.

Kind regards, 

on behalf of

Dr. Zhihong (Arry) Yao 

Academic Editor

PLOS ONE